# The efficacy of conditioned medium released by tonsil-derived mesenchymal stem cells in a chronic murine colitis model

Ko Eun Lee[1], Sung-Ae Jung[1]*, Yang-Hee Joo[1], Eun Mi Song[1], Chang Mo Moon[1], Seong-Eun Kim[1], Inho Jo[2]

**1** Department of Internal Medicine, College of Medicine, Ewha Womans University, Seoul, South Korea,
**2** Department of Molecular Medicine, College of Medicine, Ewha Womans University, Seoul, South Korea

* jassa@ewha.ac.kr

**Data Availability Statement:** All relevant data are within the article and its Supporting Information files.

## Abstract

Tonsil-derived mesenchymal stem cells (TMSC) have characteristics of MSC and have many advantages. In our previous studies, intraperitoneal (IP) injection of TMSC in acute and chronic colitis mouse models improved the disease activity index, colon length, and the expression levels of proinflammatory cytokines. However, TMSC were not observed to migrate to the inflammation site in the intestine. The aim of this study was to verify the therapeutic effect of conditioned medium (CM) released by TMSC (TMSC-CM) in a mouse model of dextran sulfate sodium (DSS)-induced chronic colitis. TMSC-CM was used after seeding $5 \times 10^5$ cells onto a 100 mm dish and culturing for 5–7 days. TMSC-CM was concentrated (TMSC-CM-conc) by three times using a 100 kDa cut-off centrifugal filter. Seven-week-old C57BL/6 mice were randomly assigned to the following 5 groups: 1) normal, 2) colitis, 3) TMSC, 4) TMSC-CM, and 5) TMSC-CM-conc. Chronic colitis was induced by continuous oral administration of 1.5% dextran sulfate sodium (DSS) for 5 days, followed by 5 additional days of tap water feeding. This cycle was repeated two more times (total 30 days). Phosphate buffered saline (in the colitis group), TMSC, TMSC-CM, and TMSC-CM-conc were injected via IP route 4, 4, 12, and 4 times, respectively. Reduction of disease activity index, weight gain, recovery of colon length, and decreased in the expression level of the proinflammatory cytokines, interleukin (IL)-1β, IL-6, and IL-17 were observed at day 30 in the treatment groups, compared to control. However, histological colitis scoring and the expression level of tumor necrosis factor α and IL-10 did not differ significantly between each group. TMSC-CM showed an equivalent effect to TMSC related to the improvement of inflammation in the chronic colitis mouse model. The data obtained support the use of TMSC-CM to treat inflammatory bowel disease without any cell transplantation.

## Introduction

Inflammatory bowel disease (IBD) is a chronic gastrointestinal disease characterized by wax and wane inflammation. The prevalence of IBD is gradually increasing in Korea as well as in

**Funding:** This research was supported by Basic Science Research Program through the National Research Foundation of Korea (NRF) funded by the Ministry of Education (grant number, NRF 2016R1A2B4006972 and NRF 2019R1A2C1002526) to SJ.

**Competing interests:** The authors have declared that no competing interests exist.

western countries [1–3]. The pathogenesis of IBD is not clearly understood. However, it is generally agreed that genetic factors, environmental factors, and immunoregulatory factors such as excessive immune reaction to the intestinal microbiome are involved [4, 5]. For the treatment of IBD, biological, anti-cytokine, and anti-adhesion molecule medications are being developed in addition to conventional therapies that include 5-aminosalicylic acid, corticosteroids, and immunomodulators. However, treatment outcome is less than ideal in 20 to 30% of patients [6, 7]. Recently stem cell therapy has been recognized as a potential new therapeutic method for such intractable diseases [8].

Mesenchymal stem cells (MSC) are present in various types of tissues. They are pluripotent and can differentiate to many kinds of mesodermal cells such as adipocytes, chondrocytes, and osteocytes [9–12]. There have been many attempts to use MSC for the treatment of many types of diseases, based on their ability to suppress immunocyte proliferation and to control inflammation, because of their secretion of various cytokines [9, 13]. A number of human and animal studies have demonstrated the anti-inflammatory effects of MSC in IBD [8, 14, 15].

Bone marrow and adipose tissues are major sources of MSC. Recently, tonsil-derived mesenchymal stem cells (TMSC) were reported to have characteristics of MSC [16–18]. TMSC has several advantages compared with other MSC. They are easily acquired from resected and discarded tissues after tonsillectomy, a commonly performed surgical procedure in children, and the doubling time of TMSC is short, possibly owing to the young age of the donor. Secondly, TMSC acquired from many donors tend to unify and grow well without the domination by a certain cell line, and have pronounced differentiation capacity and immune modulatory activity [17]. In addition, the characteristics of stem cells are well conserved and are not, damaged after freezing and de-freezing. Therefore, TMSC may serve as a source for future stem cell banks and may also be easily used clinically.

In our previous studies, the intraperitoneal (IP) administration of TMSC in dextran sulfate sodium (DSS)-induced acute and chronic colitis animal models demonstrated improvements in disease activity index (DAI), colon length, and the expression of pro-inflammatory cytokines [19, 20]. To trace the localization of the TMSC *in vivo*, TMSC tagged with PKH26 red fluorescent cell linker were administered to normal mice and mice with acute colitis. Fluorescence microscopy examination of the colon tissues acquired after 5 days did not reveal any red TMSC-PKH26 cells [19]. Therefore, it was presumed that the TMSC did not migrate directly to the inflammation site of the intestine, with the results explained by the paracrine effect of the TMSC.

This study aimed to evaluate the therapeutic effect of the conditioned medium (CM) released by TMSC (TMSC-CM) in DSS-induced chronic murine colitis models. TMSC-CM is more feasible for preparation and application than TMSC themselves. We hypothesized that TMSC-CM will exert therapeutic effects similar to those of TMSC upon IP administration.

## Materials and methods

### Isolation of TMSC

TMSC developed in 'Ewha Tonsil-derived Mesenchymal Stem Cell Research Center (ETSRC)' were used. The tonsil tissue was acquired from the patients who agreed to provide resected tissues from their tonsillectomy surgeries after informed consent. The process was approved by the Institutional Review Board of Ewha Womans University Mokdong Hospital (ECT 11-53-02). The palatine tonsil tissue was acquired immediately after surgery and washed 5 times with phosphate buffered saline (PBS), followed by being minced. They were then digested for 30 min at 37°C in a Roswell Park Memorial Institute 1640 (RPMI-1640) medium (Invitrogen, Carlsbad, CA, USA) with 210 U/mL collagenase type 1 (Invitrogen) and 10 μg/mL DNase

(Sigma-Aldrich, St. Louis, MO, USA). After being passed through a cell strainer, the cells were washed twice with 20% human serum and 10% human serum in RPMI 1640 medium. Monocytes were isolated by Ficoll-Paque (GE Healthcare, Buckinghamshire, UK) density gradient centrifugation. The cells were cultured in low-glucose Dulbecco's modified Eagle's medium (DMEM) (Welgene, Daegu, South Korea) containing antibiotics and 10% heat-inactivated fetal bovine serum (FBS) (Capricorn, Ebsdorfergrund, Germany), until they attained a cell density of $5 \times 10^5$ cells/well, and non-adherent cells were removed after 24 h. TMSC of passage 5–7 were used for the experiment.

## Preparation of TMSC conditioned medium

TMSC conditioned medium (TMSC-CM) was acquired after seeding $5 \times 10^5$ TMSC (in 10ml low-glucose DMEM containing 10% FBS) in a 100 mm dish, harvesting the cells for 5–7 days (total $2 \times 10^6$ cells), and removing the cell debris by centrifugation at 3,800 $\times g$ for 5 min at 4˚C. The medium was then collected, filtered through a 0.2 μm filter, and then stored at 4˚C.

TMSC-CM was triple-concentrated (TMSC-CM-conc) using a 100 kDa cut-off centrifugal filter (Amicon Ultra, Merck Millipore, Burlington, MA, USA) at 3,800 $\times g$ for 5 min at 4˚C, and the medium that was heavier than 100 kDa was collected from the upper tube.

## *In vitro* immunosuppression assay

Immunosuppressive effects of TMSC-CM were confirmed by splenocyte immunosuppression assay. Splenocyte isolation ($1 \times 10^6$ cells) was performed from the spleen of a C57BL/6 mouse. Splenocytes were stimulated with the following mitogens; 5 μg/mL of lipopolysaccharide (LPS) (Sigma-Aldrich) for B-cell activation, or 20 ng/mL of phorbol 12-myristate 13-acetate (PMA) (Sigma-Aldrich) plus 1 μg/mL of ionomycin (Sigma-Aldrich) for T-cell activation. The activated splenocytes were cultured in TMSC-CM or TMSC-CM-conc, or co-cultured with TMSC for 24 h. After 24 h, proliferation of splenocytes was measured with an ELISA reader, using the Cell Counting Kit (CCK)-8 assay kit (Dojindo Molecular Technologies, Rockville, MD, USA) at 450 nm.

## Analysis of the TMSC-CM components

To identify proteins from the CM, 1 mL of the basal medium, low-glucose DMEM (10% FBS), and 1 mL of the TMSC-CM were each loaded onto cytokine and angiogenesis array membrane (R&D systems, Inc., Minneapolis, Minnesota, USA). After blocking the array membrane with blocking buffer for 1 h and membrane washing, the TMSC-CM and array detection antibody cocktail was mixed and added to the blocked membrane followed by overnight shaking incubation at 4˚C. After washing, streptavidin-HRP buffer was added to the membrane, and incubation was performed for 30 min. Following another washing, Chemi Reagent Mixture was added to the membrane for reaction at room temperature for 1 min and measured using LAS-300 system (Fujifilm, Tokyo, Japan). Dot density was analyzed using Multi Gauge 3.0 software. Anti-inflammatory and pro-inflammatory cytokine levels were measured using cytokine array, and growth factor levels were measured using angiogenesis array.

## Development of a chronic colitis mouse model

The animal models used for the experiment were seven-week-old C57BL/6 male mice (Orient Bio Co., Ltd., Sungnam, Gyeonggi, Korea) with an average weight of 20–22 g. The mice were allowed 7 days of adaptation period at the facility of the Ewha Womans University Medical Research Institute prior to the experiment. The environment was set to standardized

environment for study animals. Day time and night time were provided at 12 h intervals, and temperature (23 ± 2°C) and humidity (45–55%) were set to appropriate levels. This study was approved by the Ethics Committee for Animal Research of Ewha Womans University (ESM 15–0312).

Chronic colitis was induced by oral administration of 1.5% DSS (molecular weight 36 ± 50 kDa, MP biochemical, Irvine, CA, USA) for 5 days continuously followed by an additional 5 days of tap water feeding; a total of 3 such cycles (total 30 days) was performed.

### *In vivo* experimental design

Mice were randomly assigned into 5 groups: 1) the normal control, 2) colitis control, 3) TMSC injection, 4) TMSC-CM injection, and 5) TMSC-CM-conc injection groups.

Normal control group mice were provided with standardized water and food without any pre-treatment. In the colitis control group and three other treatment groups, chronic colitis was induced with 1.5% DSS. For the colitis control group, PBS was IP injected to match the effect of cell administration-related stress with the TMSC group. Mice in the colitis control group received four IP injections of 500 μL of PBS, those in the TMSC group received four IP injections of $1\times10^6$ TMSC/500 μL. The TMSC-CM group received 12 IP injections of 500 μL of TMSC-CM (equivalent to $1\times10^5$ TMSC) in order to match the amount of TMSC with the TMSC group. As the number of injections increases, the injection-related stress of the mouse increases, therefore, the same number of injections could be used to reduce the bias. The TMSC-CM-conc group, which received four IP injection of 500 μL of TMSC-CM-conc (equivalent to $3\times10^5$ TMSC), was designed in order to match the number of injections with that of the TMSC group (Fig 1).

### Assessment of the effect of TMSC and TMSC-CM

The severity of colitis in each mouse was assessed by measuring their DAI, body weight change, colon length, histologic grading, and cytokine levels. For each group, weight change,

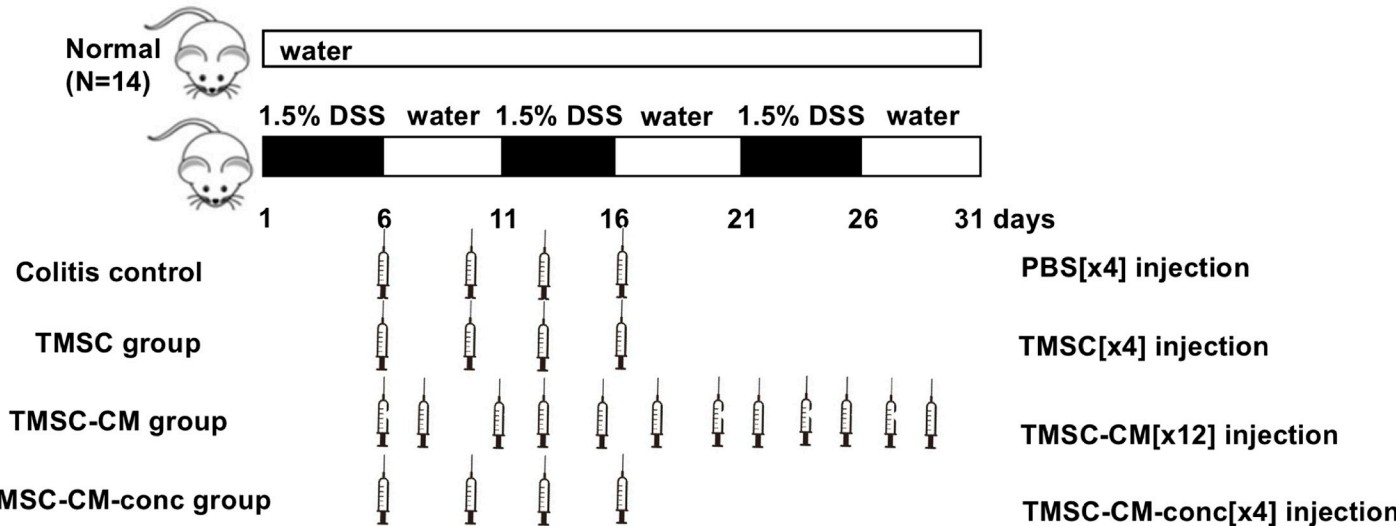

**Fig 1. Experimental design.** Mice were randomly assigned into 5 groups: the normal control, colitis control, TMSC injection, TMSC-CM injection, TMSC-CM-conc injection groups. Mice in the colitis control group received 4 IP injections of PBS, those in the TMSC group received 4 IP injections of TMSC, those in the TMSC-CM group received 12 IP injections of TMSC-CM, and those in the TMSC-CM-conc group received 4 IP injections of triple-concentrated TMSC-CM. DSS, dextran sulfate sodium; PBS, phosphate buffered saline.

stool consistency, and occult or grossly observed blood from stool or anus were checked daily and the DAI was assessed (S1 Table) [21]. On day 31 of the experiment, mice were sacrificed with $CO_2$ inhalation, and colon specimens were acquired from the proximal and distal parts of the dissected colon and fixed with 10% formalin, followed by paraffin sectioning and hematoxylin-eosin (H&E) staining. Two specimens from each proximal and distal colon were analyzed using a previously reported histologic colitis scoring system (S2 Table) [22].

The colon specimen was frozen to -70˚C in the cryogenic freezer until analysis. One milliliter of TRIzol Reagent (Ambion, Life Technologies, Carlsbad, CA, USA) was added to the colon specimen, and after being minced, 200 μL of chloroform (Sigma) was added, followed by vortexing. After leaving the sample at the room temperature for 2 to 3 min, it was centrifuged at 13,000 ×g (4˚C) for 10 min. The supernatants were added into a new tube and the same amount of isopropanol (Sigma) was added; after mixing by inversion, the sample was left at room temperature for 5 min, and then centrifuged again at 13,000 ×g (4˚C) for 10 min. The cell pellet was washed with 75% ethanol for 2–3 times and dried at room temperature. Nuclease-free water was then added, and the RNA concentration was measured with Nabi (nanodrop) (Micro Digital, Korea). Then, 2 μg of RNA and 0.5 μg of oligo dT primer were mixed, and the sample was left at 70˚C for 10 min. Next, 200 units of Molony Murine Leukemia Virus Reverse Transcriptase (M-MLV RT) (Promega, Fitchburg, WI, USA), 25 units of rRNasin Ribonuclease inhibitor (Promega), 5 X RT buffer and 2 mM of dNTP were added, and the final volume was set to 25 μL by adding an appropriate amount of nuclease-free water. They were treated at 42˚C for 60 min, and 95˚C for 5 min, and then stored at 4˚C. Quantstudio 3 real-time polymerase chain reaction (PCR) system (Applied Biosystems, Waltham, MA, USA) was used for analysis, using 0.1 μg of the synthesized cDNA as a template for the 2X Power SYBR Green PCR Master mix (Applied Biosystems) and each primer set. The primers used in this study were made by Macrogen (Korea). Pro-inflammatory cytokines, interleukin (IL)-1β, IL-6, tumor necrosis factor α (TNFα), IL-17, and the anti-inflammatory cytokine IL-10 levels were measured; each primer sequence is shown in Table 1. Each PCR was performed after 10 min of pre-denaturation at 95˚C, 15 seconds at 95˚C and 1 min at 60˚C; this was repeated 40 times. After the PCR was complete, a melting curve was drawn to check the accuracy of gene amplification. For the internal compensation of gene expression level, the house keeping gene, glyceraldehyde-3-phosphate dehydrogenase (GAPDH), was also used, and relative gene expression levels were presented as $2^{-\Delta\Delta Ct}$ values.

**Table 1. Primer sequences of the Reverse Transcription Polymerase Chain Reaction (real time-PCR).**

| Primer | | Sequence |
|---|---|---|
| IL-1β | Forward | 5'-GAGCCCATCCTCTGTGACTC-3' |
| | Reverse | 5'-TCCATTGAGGTGGAGAGCTT-3' |
| IL-6 | Forward | 5'-CCGGAGAGGAGACTTCACAG-3' |
| | Reverse | 5'-TCCACGATTTCCCAGAGAAC-3' |
| IL-17 | Forward | 5'-TCCCTCTGTGATCTGGGAAG-3' |
| | Reverse | 5'-CTCGACCCTGAAAGTGAAGG-3' |
| TNFα | Forward | 5'-ACGGCATGGATCTCAAAGAC-3' |
| | Reverse | 5'-AGATAGCAAATCGGCTGACG-3' |
| GAPDH | Forward | 5'-TGATGACATCAAGAAGGTGGTGAAG-3' |
| | Reverse | 5'-TCCTTGGAGGCCATGTGGGCCAT-3' |

IL, interleukin; TNF, tumor necrosis factor; GAPDH, glyceraldehyde-3-phosphate dehydrogenase

### Statistical analysis

For each of the analyzed parameters, values were presented as mean ± standard deviation (SD). Analysis of variance (ANOVA) test followed by post-hoc analysis using Tukey's comparison test was used to compare the results of the control group with those of the TMSC, TMSC-CM, and TMSC-CM-conc group ($*p < 0.05$, $**p < 0.01$, and $***p < 0.001$, respectively). All statistical analyses were performed using GraphPad Prism software version 6 (GraphPad Software, Inc., La Jolla, CA, USA); a $p$-value less than 0.05 was considered statistically significant.

## Results

### Effects of TMSC and TMSC-CM on splenocyte proliferation

The rate of proliferation of splenocytes was significantly reduced by the number of TMSC and the concentration of TMSC-CM. Splenocytes stimulated by LPS or PMA/ionomycin showed a decreased proliferation rate (fold change) when co-cultured with TMSC, and cultured in TMSC-CM; the proliferation rates were 7.30 ± 0.43, 5.77 ± 0.56, 4.88 ± 0.80, 4.28 ± 0.22, 5.56 ± 0.18, and 4.45 ± 0.87 in LPS, LPS + $5\times10^3$ TMSC, LPS + $5\times10^4$ TMSC, LPS + $5\times10^5$ TMSC, LPS + TMSC-CM, and LPS + TMSC-CM-conc, respectively ($p < 0.0001$, ANOVA); 6.49 ± 0.13, 4.71 ± 0.06, 4.34 ± 0.83, 4.30 ± 0.30, 2.27 ± 0.33, and 2.13 ± 0.47 (stimulated by PMA/ionomycin) ($p < 0.0001$, ANOVA) (Fig 2).

### Anti-inflammatory protein expression of TMSC-CM

Following the administration of TMSC-CM, cytokines and growth factors were highly expressed compared to the use of basal medium (Fig 3(A)). Proteins that were highly expressed following treatment with TMSC-CM included growth factors such as hepatocyte growth factor (HGF), placental growth factor (PlGF), and vascular endothelial growth factor (VEGF), tissue repair-related proteins including thrombospondin-1, urokinase-type plasminogen activator

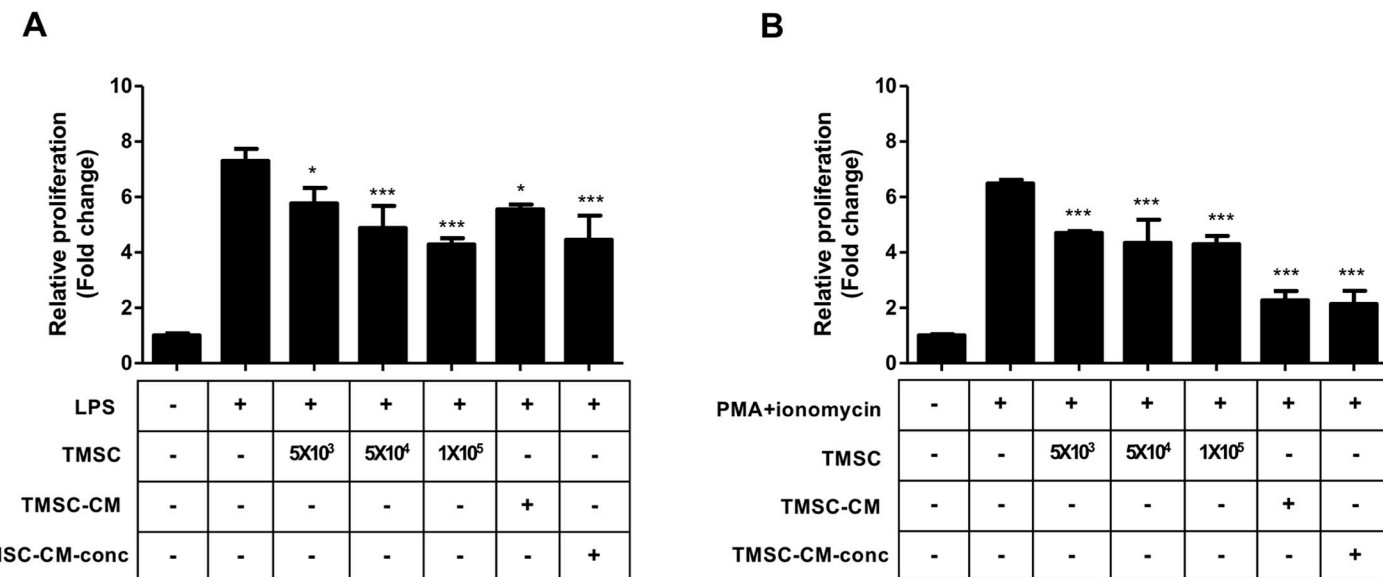

**Fig 2. Splenocyte immunosuppression assay.** The splenocyte stimulated by LPS (A) or PMA/ionomycin (B) shows decreased proliferation rate when co-cultured with TMSC and cultured in TMSC-CM. Proliferation was significantly reduced by the number of TMSC and the concentration of TMSC-CM. (mean with standard deviation, $*p < 0.05$, $***p < 0.001$) LPS, lipopolysaccharide; PMA, phorbol 12-myristate 13-acetate.

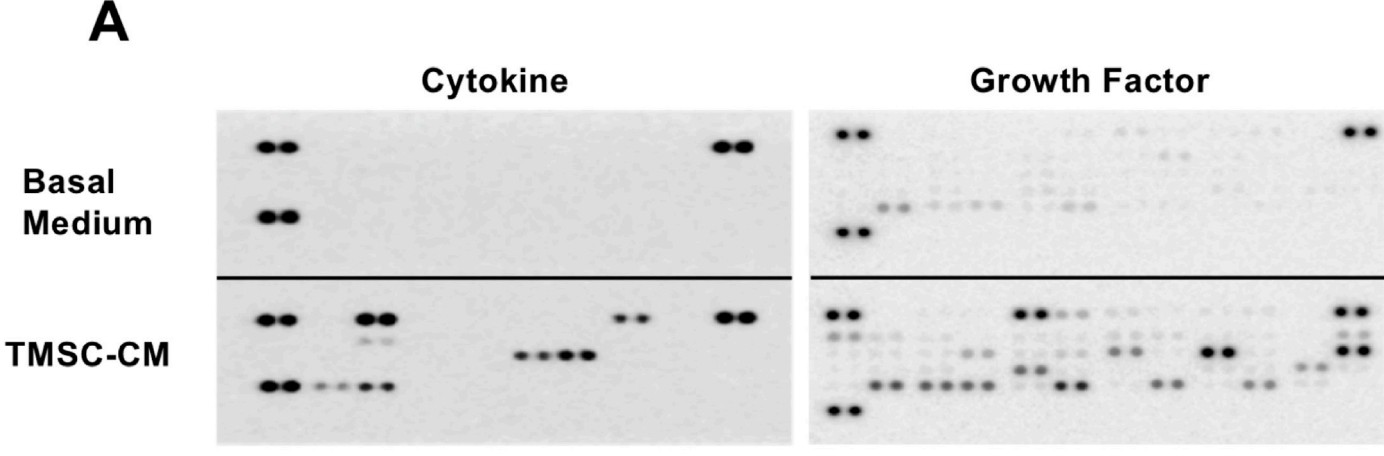

**Fig 3. Cytokine and angiogenesis array.** (A) Basal medium and TMSC-CM are applied to each array membrane. (B) When administered with TMSC-CM, growth factors, tissue repair, anti-inflammatory, and anti-apoptosis proteins were highly expressed compared to when the basal medium was applied. (mean with standard deviation) O.D., optical density; HGF, hepatocyte growth factor; PIGF, placental growth factor; VEGF, vascular endothelial growth factor; FGF, fibroblast growth factor; HB-EGF, heparin binding epidermal growth factor-like growth factor; uPA, urokinase-type plasminogen activator; MCP, monocyte chemoattractant protein; CXCL, C-X-C Motif chemokine ligand; SDF, stromal cell-derived factor; IL, interleukin; GM-CSF, granulocyte-macrophage colony-stimulating factor; TIMP, tissue inhibitor of metalloproteinase; IGFBP, insulin-like growth factor-binding protein.

(uPA), and monocyte chemoattractant protein (MCP)-1, the anti-inflammatory protein transforming growth factor (TGF)-β, and the anti-apoptosis protein tissue inhibitor of metalloproteinase (TIMP)-1 (Fig 3(B)).

## Protective effect of TMSC and TMSC-CM on colonic inflammation in DSS mouse model

A total of 61 male mice were randomly assigned into 5 groups: normal control (n = 14), colitis control (n = 14), TMSC group (n = 9), TMSC-CM group (n = 14), and TMSC-CM-conc group (n = 10).

On day 30 of the experiment, DAI was significantly decreased in the TMSC, TMSC-CM, and TMSC-CM-conc groups compared to that in the colitis control group ($4.44 \pm 3.91$, $1.80 \pm 0.54$, $1.54 \pm 0.69$, and $1.28 \pm 0.67$ in the colitis, TMSC, TMSC-CM, and TMSC-CM-conc group, respectively ($p = 0.0022$, ANOVA)). There were no significant differences between the DAI values for each of the treatment groups (Fig 4). The weight change (%) at day 30 for the TMSC, TMSC-CM, and TMSC-CM-conc groups was significantly increased compared to that of the colitis control group ($-2.69 \pm 19.79$, $9.25 \pm 6.53$, $11.12 \pm 12.12$, and $13.57 \pm 7.20$, $p = 0.0183$). There were no significant differences between the weight gain (%) values for each of the treatment groups (Fig 5). The recovery of colon length (cm) observed at day 30 for the TMSC, TMSC-CM, and TMSC-CM-conc groups were significantly better than the recovery in the colitis control group ($72.21 \pm 6.55$, $80.68 \pm 5.87$, $80.5 \pm 7.05$, and $81.8 \pm 5.57$, $p = 0.0014$). There were no significant differences between the colon length (cm) values for each of the treatment groups (Fig 6) (S3 Table). The survival rate of the colitis control group was 78.6% and 100% for the three treatment groups.

## Histopathological alteration

Histopathologic findings of the dissected colon specimen showed infiltration of mononuclear cells and crypt damage, but no perforation. Histologic colitis scoring assessed after H&E staining did not show any significant differences between each group ($p = 0.2933$, ANOVA) (Fig 7, S4 Table).

## Cytokine levels of colon tissues treated with TMSC or TMSC-CM

With respect to the cytokine levels of the colon tissue, the level of the IL-1β proinflammatory cytokine was significantly lower in the TMSC, TMSC-CM, and TMSC-CM-conc groups than in the colitis control group ($p = 0.0188$, ANOVA). The IL-6 level was significantly lower in the TMSC and TMSC-CM-conc groups ($p = 0.0103$, ANOVA), and the IL-17 level was significantly lower in the TMSC-CM and TMSC-CM-conc groups ($p = 0.0456$, ANOVA) than in the colitis control group. There were no significant differences between the cytokine levels in each of the treatment groups. There were no significant differences between the expression levels of TNFα ($p = 0.1611$, ANOVA) and IL-10 ($p = 0.4506$, ANOVA) in each group (Fig 8, S5 Table).

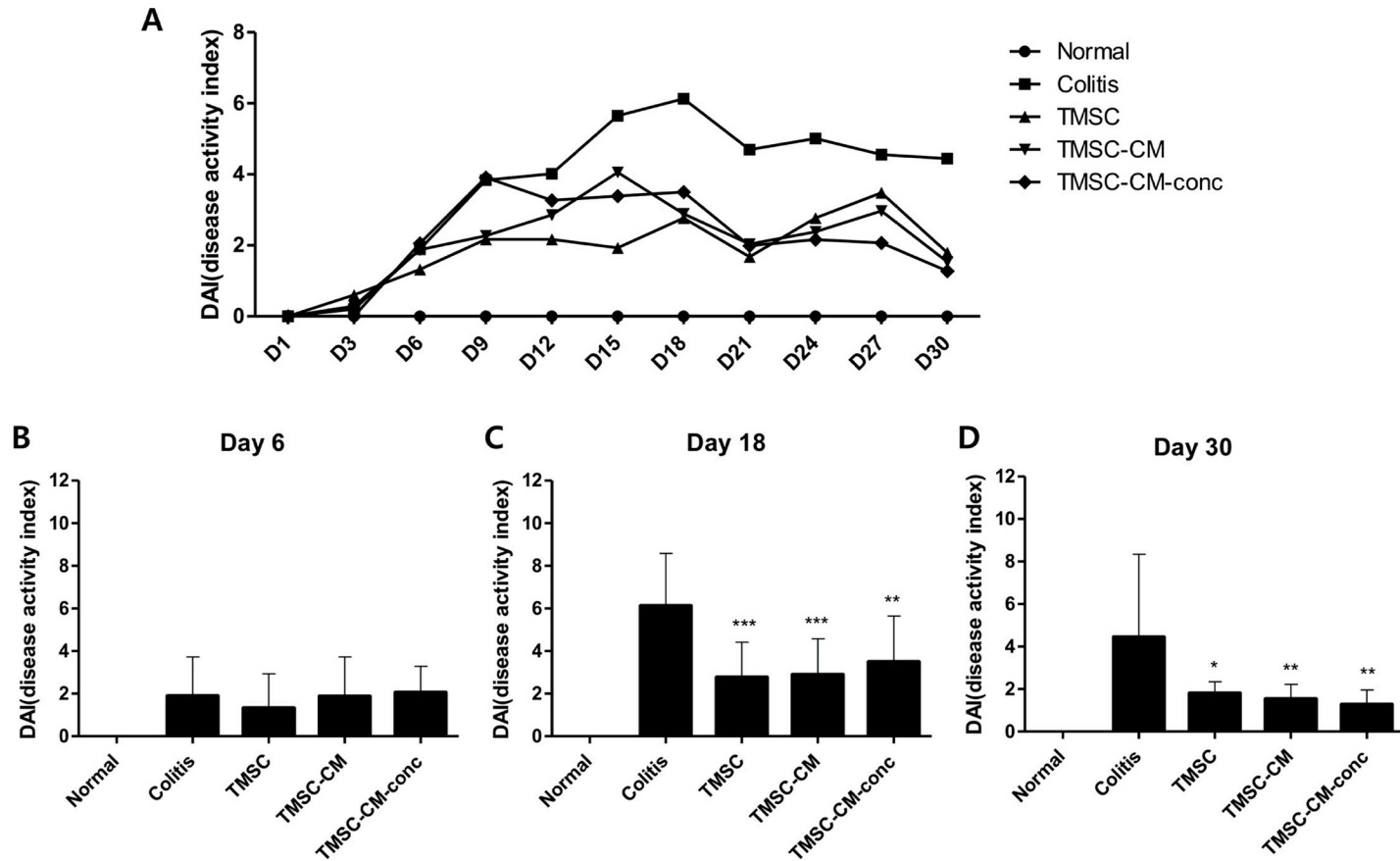

**Fig 4. Disease activity index (DAI).** DAI was significantly decreased in TMSC, TMSC-CM, and TMSC-CM-conc group compared to the colitis control group at the 18th, 30th day of experiment. (mean with standard deviation, $^*p < 0.05$, $^{**}p < 0.01$, $^{***}p < 0.001$).

## Discussion

To the best of our knowledge, this is the first study to evaluate the therapeutic effect of TMSC-CM in a chronic colitis mouse model. It is clinically meaningful that the effect of TMSC and TMSC-CM are similar, because TMSC-CM is easier to prepare and administer to patients. In addition, the amount of TMSC-CM used in this study included fewer cells compared to the TMSC group, whereas the therapeutic effects were similar, suggesting that the therapeutic effect of TMSC-CM is clinically significant.

Many previous reports have reported the therapeutic effects of MSC for colitis. When human adipose-derived MSC (ASC) were IP injected in the DSS-induced acute and chronic colitis mouse models, histopathological severity of colitis, weight loss, diarrhea, inflammation, and survival rates improved, and ASC were reported to decrease inflammatory cytokine levels and correlated with the induction of IL-10-secreting T regulatory cells [23]. Another study demonstrated that the IP injection of nucleotide-binding oligomerization domain 2 (NOD2)-activated human umbilical cord blood (UCB) MSC into DSS or trinitrobenzene sulfonic acid (TNBS)-induced colitis mouse models resulted in the decreased severity of colitis and increased anti-inflammatory responses such as expression of IL-10 and T regulatory cells [15]. The IP injection of bone marrow-derived MSC (BM-MSC) into DSS-induced acute colitis models resulted in the reduction of colitis and recovery of body weight associated with TNFα-stimulated gene-6, an important immunoregulatory molecule [24]. In the present study, the

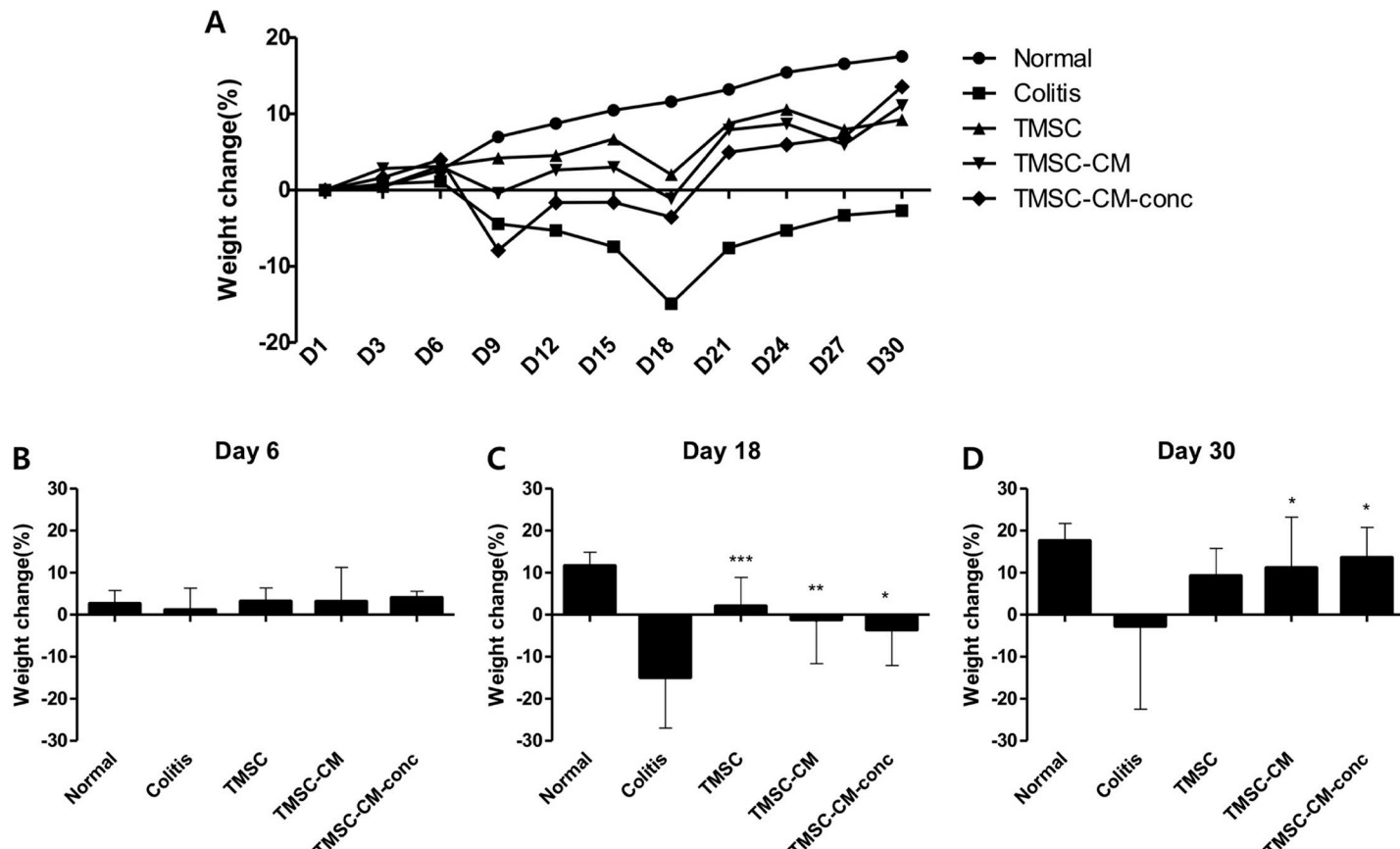

**Fig 5. Weight recovery.** The weight recovery at the 18th, 30th day of experiment was significantly greater in TMSC, TMSC-CM, and TMSC-CM-conc groups compared to that in colitis control group. (mean with standard deviation, $^*p < 0.05$, $^{**}p < 0.01$, $^{***}p < 0.001$).

TMSC injection group showed improvements in severity of colitis, weight loss, and colon length compared to the control group, and levels of proinflammatory cytokines including IL-1β and IL-6 were low, similar to the results of previous studies which assessed the effect of MSC.

The effect of MSC-CM has not been investigated thoroughly compared to the effect of the MSC injection itself. In a previous study using an acute colitis model, human BM-MSC and BM-MSC-CM were each administered by enema to TNBS-induced acute colitis guinea pig models with enteric neuropathy and motility dysfunction. Both treatment groups showed improvements in weight loss and gross morphological, and histological colon damage, with the prevention of loss of myenteric neuron and nerve process damage, compared to the colitis control group. There was no significant difference between BM-MSC- and BM-MSC-CM-treated groups [25]. In another study using human amnion-derived MSC, MSC were intravenously injected and MSC-CM was administered via enema to TNBS-induced acute colitis rat models. Both groups showed significant improvements in the endoscopic score and decreased infiltration of neutrophils, monocytes, macrophages, TNF-α, C-X-C Motif chemokine ligand 1 (CXCL1), and C-C motif chemokine ligand 2 (CCL2) [26]. In the present study, we evaluated the effect of TMSC-CM in a chronic colitis model induced by DSS, which is more similar model to human IBD, which features chronic relapsing and remitting episodes.

The similar effects of MSC and MSC-CM support the concept that MSC themselves does not migrate and act on the target, but rather that the substances that the cells secrete such as

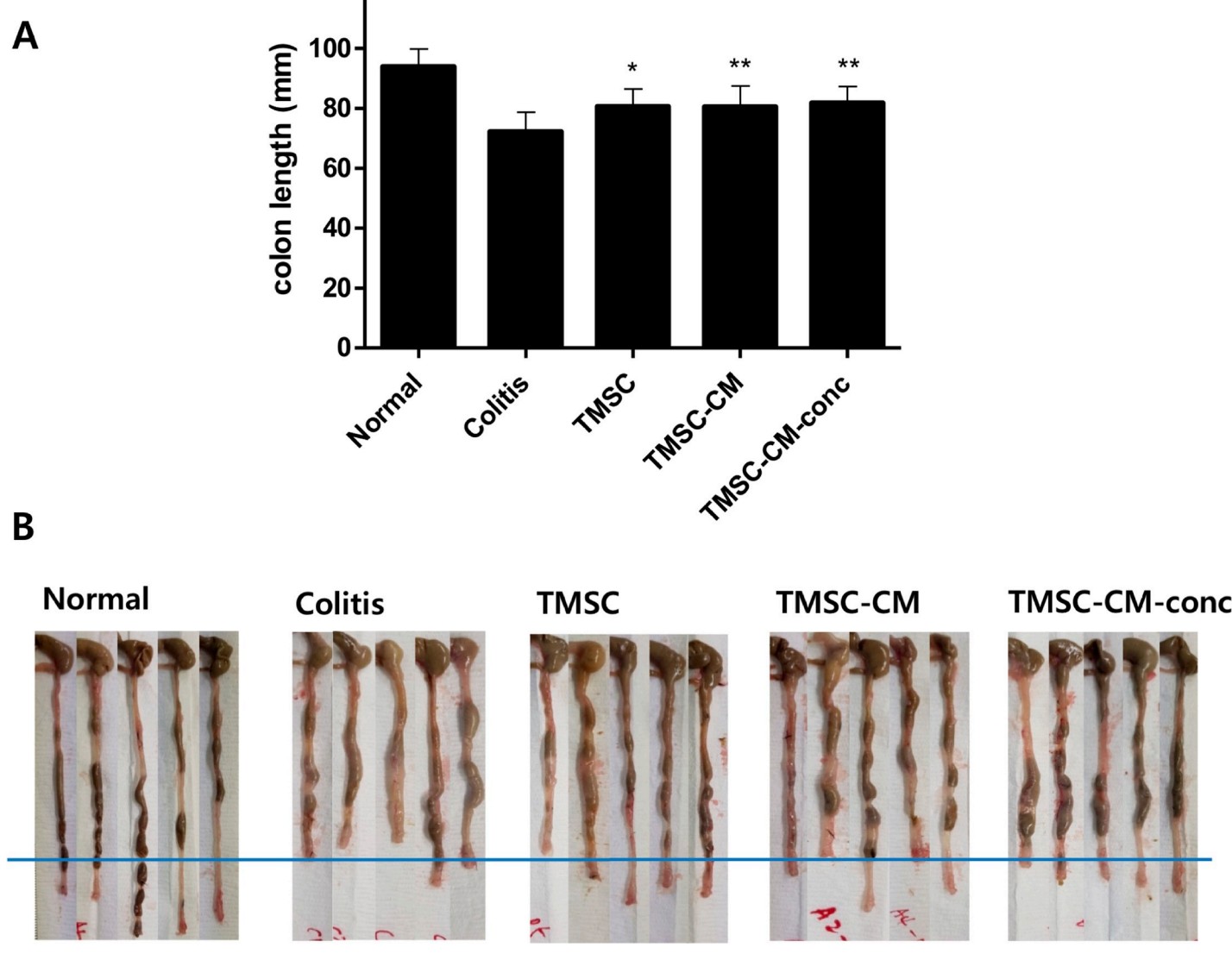

**Fig 6. Colon length.** (A) Colon length at the 30th day of experiment was significantly increased in TMSC, TMSC-CM, and TMSC-CM-conc group compared to the colitis control group. (mean with standard deviation, $^*p < 0.05$, $^{**}p < 0.01$) (B) Resected colons of each group are shown.

extracellular vesicles (EVs) or soluble proteins, contribute to the treatment effect. In a study in which BM-MSC were IP injected into a DSS-induced acute colitis animal model and BM-MSC were simultaneously tracked, aggregation was observed mostly with macrophages, B cells, and T cells, and were located in the peritoneal cavity, whereas fewer than 1% reached the inflamed colon [24]. MSC did not migrate directly to the colon, but decreased colitis was observed. It has also been reported that when MSC-CM is applied to hypoxic-damaged cardiomyocytes, it protects cardiomyocytes by the paracrine effect, which interferes with mitochondria-mediated apoptosis [27]. However, some studies have demonstrated that the injected MSC move directly towards the target. In a study conducted in a rat model with calvarial bone defect, fluorescent-labeled rat MSC were injected into the caudal vein and graft material made from BM-MSC-CM was applied to the lesion. The MSC gradually migrated to the defect site [28]. In an acute pancreatitis rat model, fluorescent-labeled BM-MSC were injected and it migrated to the

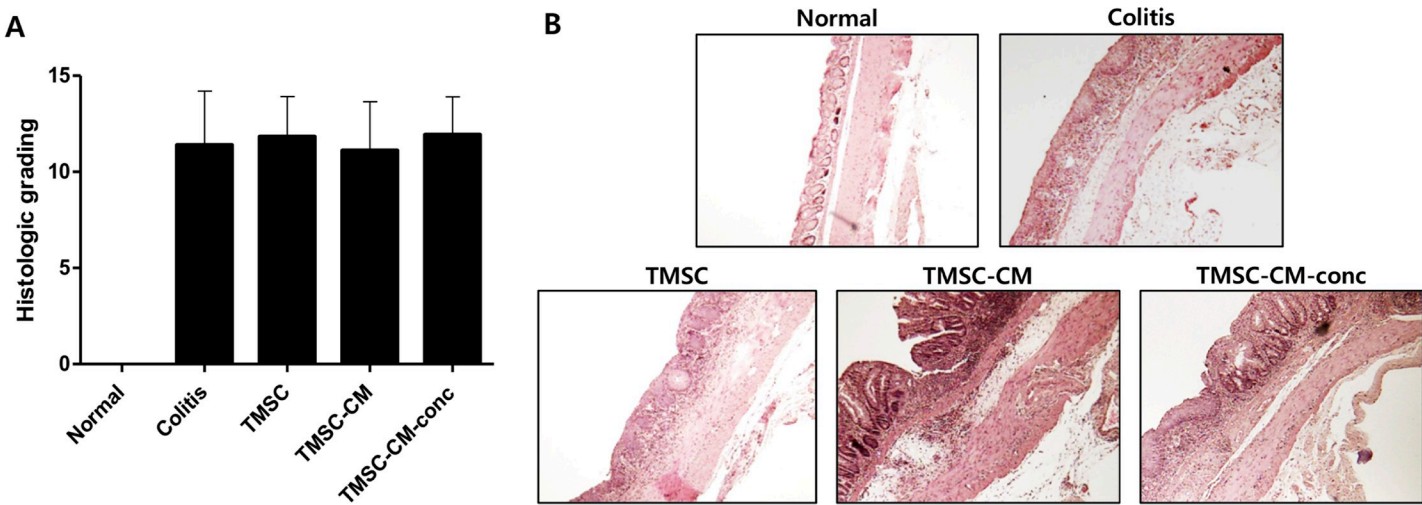

**Fig 7. Histopathological alteration.** (A) Histologic colitis scoring did not show significant difference between each of groups. (B) Histopathologic findings of distal colon of each group are shown. (Hematoxylin-eosin (H&E) stain, 100x).

**Fig 8. Cytokine levels of colon tissue.** The level of IL-1β was significantly lower in TMSC, TMSC-CM, and TMSC-CM-conc groups compared to the colitis control group. (mean with standard deviation, $^*p < 0.05$, $^{**}p < 0.01$) IL, interleukin; TNF, tumor necrosis factor.

damaged site of the pancreas [29]. Therefore, further studies are necessary to elucidate MSC migration according to the disease model.

Recently, MSC-derived EVs have been reported to be effective through cell-to-cell communication, cell signaling, and altering cell or tissue metabolism in the body [30]. In addition, several reports have suggested that EVs may be used for treatment as a substitute for cell therapy [31–34]. Intramyocardial injection of EVs sourced from human BM-MSC into rats with acute myocardial infarction reduced the infarct size and improved cardiac function [35]. Intravenous (IV) injection of exosomes obtained from human UCB MSC into acute lung injury mice resulted in the suppression of hypoxic inflammation [36]. In a skin second-degree burn wound rat model, human UCB MSC exosomes were injected subcutaneously and resulted in wound healing due to increased re-epithelialization [37]. Periocular injection of MSC exosomes in an autoimmune uveitis rat model reduced infiltration in T cells and other inflammatory cells, and eventually improved uveitis [38]. In this study, because TMSC-CM were filtered using a 0.2 μm filter, only EVs with diameters less than 200 nm were extracted from the TMSC-CM. After centrifugation with a 100 kDa cut-off filter, only the upper tube substrates heavier than 100 kDa were selected, which can be converted to particles with a diameter of 10 nm or larger. Since the diameter of the exosome generally ranges from 30 to 100 nm [39], the particle size range of 10 to 200 nm in this study would have contained a slightly wider range of exosomes. Therefore, the exosomes in TMSC-CM may have induced therapeutic effects in chronic colitis, which is consistent with results of previous studies. However, in our previous attempt, the lower tube materials weighing less than 100 kDa which were produced after co-culturing with PMA/ionomycin-stimulated splenocytes also showed therapeutic effects. Therefore, it is necessary to conduct repetitive experiments using EVs in each weight range to specifically evaluate their influence.

Collectively, these results support the view that administering MSC themselves and only MSC-CM results in equivalent therapeutic effects, which may support the application of MSC-CM over cell transplantation. This application is desirable, because it would be easier to perform and would also be accompanied with fewer side effects. Additionally, MSC-CM was simpler to prepare and store than MSC during the experiment; therefore, MSC-CM would be more feasible for clinical applications. In contrast to other MSC, TMSC differentiate into the endoderm as well as the mesoderm, and are readily and inexpensively available, as they can be acquired from tonsillectomy surgeries, which are commonly performed in children. They also have shorter doubling time because of the young age of the donors. These factors may make TMSC superior to other MSC in many aspects [18]. Therefore, the development and application of a TMSC-CM-derived therapeutic agent for IBD treatment would be highly effective and clinically feasible.

There were several limitations to this study. First, TMSC and TMSC-CM were injected through the IP route. For clinical applications, the drug should be delivered either intravenously or transanally. In a previous attempt by us, TMSC administration via IV injection and transanal enema were also attempted. These were not as effective as IP injection, possibly owing to the difficulty of IV administration through the tail vein of the mouse, which creates a stressful environment for the mouse. When transanal enema was attempted, most of the liquid leaked out of the anus just after the enema, which resulted in insignificant effects. Secondly, the injected amount of TMSC or TMSC-CM for each injection was high. Generally, the appropriate injection amount for IP injection in a mouse model is 20 to 80 mL/kg [40]; in this study the average weight of the mouse was 20 to 22g. Therefore, the appropriate injection amount was 400 to 1600 μL per injection. Considering that weight loss represents the induction of chronic colitis, 500 μL was injected. However, this amount is excessive when considering human injection dosage. Therefore, it is necessary to concentrate the injectate as much as

possible and reduce the amount in single doses, within the range that the solution does not solidify. Thirdly, the histological colitis scoring system was used for histologically comparing the H&E-stained specimen, which showed no significant findings. The scores obtained using the histological scoring system showed a significant difference between the treatment and colitis control groups in a previous study conducted using the acute colitis mouse model, in which colitis was induced for 7 days. However, in the current study using the chronic colitis model, in which colitis was induced for 30 days, it seems that the healing was not reflected in the scoring system because of the fibrotic change, resulting in no significant difference between the groups. Additional analysis regarding other histologic scoring systems should be performed. In addition, other staining methods such as Masson's trichrome staining should be performed to evaluate the distribution of the collagen fibers, which is related to chronic inflammation and fibrosis.

## Conclusion

TMSC-CM showed effects that were equivalent to those of TMSC on reducing inflammation in a DSS-induced chronic colitis model. Therefore, it is expected that administration of TMSC-CM, and not the TMSC themselves, may exert a therapeutic effect in IBD. Preparing and storing of TMSC-CM is clinically more feasible than compared to TMSC. Therefore, TMSC-CM could serve as a good therapeutic agent to treat IBD.

## Supporting information

**S1 Table. Disease Activity Index (DAI) scoring system.**
(DOCX)

**S2 Table. Histologic colitis scoring system.**
(DOCX)

**S3 Table. Disease activity index (DAI), weight change, and colon length at the 30th day of experiment.**
(DOCX)

**S4 Table. Histologic colitis scoring at the 30th day of experiment.**
(DOCX)

**S5 Table. Level of cytokine at the 30th day of experiment.**
(DOCX)

## Author Contributions

**Conceptualization:** Sung-Ae Jung, Eun Mi Song, Seong-Eun Kim, Inho Jo.

**Data curation:** Ko Eun Lee, Yang-Hee Joo.

**Formal analysis:** Ko Eun Lee, Yang-Hee Joo.

**Writing – original draft:** Ko Eun Lee, Yang-Hee Joo.

**Writing – review & editing:** Ko Eun Lee, Sung-Ae Jung, Eun Mi Song, Chang Mo Moon, Seong-Eun Kim, Inho Jo.

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
