## [Decision Letter · Decision Letter 0]

7 Aug 2019

PONE-D-19-18142

The Efficacy of Conditioned Medium Released by Tonsil-derived Mesenchymal Stem Cells in a Chronic Murine Colitis Model

PLOS ONE

Dear Prof. Jung,

Thank you for submitting your manuscript to PLOS ONE. After careful consideration, we feel that it has merit but does not fully meet PLOS ONE’s publication criteria as it currently stands. Therefore, we invite you to submit a revised version of the manuscript that addresses the points raised during the review process.

We would appreciate receiving your revised manuscript by Sep 21 2019 11:59PM. To enhance the reproducibility of your results, we recommend that if applicable you deposit your laboratory protocols in protocols.io, where a protocol can be assigned its own identifier (DOI) such that it can be cited independently in the future. For instructions see: http://journals.plos.org/plosone/s/submission-guidelines#loc-laboratory-protocols

We look forward to receiving your revised manuscript.

Kind regards,

Debabrata Banerjee, Ph.D

Academic Editor

PLOS ONE

Journal Requirements:

2. At this time, we request that you  please report additional details in your Methods section regarding animal care, as per our editorial guidelines: 1) please describe any steps taken to minimize animal suffering and distress, such as by administering analgesics, 2) please include the method of sacrifice and 3) Please describe the frequency of monitoring and the criteria used to assess animal health and well-being. Thank you for your attention to these requests.

3. PLOS ONE now requires that authors provide the original uncropped and unadjusted images underlying all blot or gel results reported in a submission’s figures or Supporting Information

files. This policy and the journal’s other requirements for blot/gel reporting and figure preparation are described in detail at https://journals.plos.org/plosone/s/figures#loc-blot-and-gel-reporting-requirements and https://journals.plos.org/plosone/s/figures#loc-preparing-figures-from-image-files. When you submit your revised manuscript, please ensure that your figures adhere fully to these guidelines and provide the original underlying images for all blot or gel data reported in your submission.  In your response letter, please note whether your blot/gel image data are in Supporting Information or posted at a public data repository, provide the repository URL if relevant, and provide specific details as to which raw blot/gel images, if any, are not available. Email us at plosone@plos.org if you have any questions.

4. Please ensure that you refer to Figure 7 in your text as, if accepted, production will need this reference to link the reader to the figure.

Additional Editor Comments:

This submission will be of interest to readers of PLos One if the authors revise the manuscript substantially

The reviewers all agreed that using TMSC conditioned medium was interesting and would be clinically feasible

They all however had several comments both scientific and stylistic and the authors are requested to address the comments in detail

The authors are encouraged to pay particular attention to comments from reviewer 1, 3 and 4 regarding

1. use of an arbitrary cut off of 100 kDa membrane for concentration

2. use of different amounts of animals in different experimental groups

3. follow up of mice for survival

4. explain better the lack of histological score improvement; too much emphasis is being placed on immune parameters

Comments from reviewer 4 should be addressed in detail

Reviewers' comments:

Reviewer's Responses to Questions

**Comments to the Author**

1. Is the manuscript technically sound, and do the data support the conclusions?

Reviewer #1: No

Reviewer #2: Yes

Reviewer #3: Yes

Reviewer #4: Yes

2. Has the statistical analysis been performed appropriately and rigorously? 

Reviewer #1: Yes

Reviewer #2: Yes

Reviewer #3: Yes

Reviewer #4: Yes

3. Have the authors made all data underlying the findings in their manuscript fully available?

Reviewer #1: Yes

Reviewer #2: Yes

Reviewer #3: Yes

Reviewer #4: Yes

4. Is the manuscript presented in an intelligible fashion and written in standard English?

Reviewer #1: Yes

Reviewer #2: Yes

Reviewer #3: No

Reviewer #4: Yes

5. Review Comments to the Author

Reviewer #1: The idea of using the secretome of MSCs to treat an inflammatory disease is appropriate. However, the data do not support the conclusion. It appears that the MSC treatment was more superior to than the secretome. The condition media used a cut off that could consist of secretome other than cytokines. The CM used was from naive MSCs. It is possible when these cells are in an inflammatory environment the secretome will be different.

Reviewer #2: In the manuscript entitled “The Efficacy of Conditioned Medium Released by Tonsil-derived Mesenchymal Stem Cells in a Chronic Murine Colitis Model” the authors describe a new approach to treat chronic colitis by means of tonsil-derived mesenchymal stem cells (TMSC) in a mouse model of chronic colitis.

The authors have shown the efficacy of tonsil-derived mesenchymal stem cells conditioned medium (TMSC-CM) in the control of this inflammatory bowel disease clinical and immunological parameters but not of the histological colitis score.

The authors underline that the preparation and storing of TMSC-CM is clinically more feasible respect to TMSC isolation and suggest that TMSC-CM therapy could serve as a pharmacological agent to treat chronic colitis.

In my opinion these data are interesting and deserve publication on PlosOne. I suggest however that the authors discuss further the lack of histological score improvement and explain better the scheme of treatments in particular the higher quantity of injections in the TMSC-CM mouse group (12 respect to 4).

Grammar and style should also be improved.

Reviewer #3: Minor English wording and syntax changes are necessary. Otherwise, it is a very good paper an provides a ready means for MSC and their products in immunotherapy

Reviewer #4: The manuscript by Ko Eun Lee, Sung-Ae Jung et al. describes the development of the approach to reduce an inflammation in the chronic colitis using a conditioning media from a human tonsil-derived mesenchymal stem cells (TMSC). Authors are already published several papers regarding the injection of TMSCs in order to treat colitis in mice models. However, they found that TMSC cells are not migrated to the inflammation site in the intestine, therefore in the current manuscript they validated the conditioned media from the TMSCs.

They found that the CM can change the disease progression like cells itself.

Authors utilized a well-established mouse model using DSS (dextran sulfate sodium) to induce the chronic and acute colitis. The assessment of the severity disease is well-established, and authors employed these parameters in they work (weight of mice, the colon length, levels of proinflammatory cytokines).

Major specific comments:

1) The rate of splenocytes was reduced (p13), but this section causes the confusion. Proliferation rates was measured in what (days)? Fig. 2 presents the folds changes. Please clarify.

2) It is understandable that the TMSC-CM causes the protective anti-inflammatory effect on the colonic inflammation. However, it is raising the question can this therapy impact on a mice immuno system causing the immunogenicity and other complications?

3) How long after this treatment mice can survive compare with the disease bearing group?

4) p.15 Fig. 4, Disease activity index (DAI) was define in the legend, please do it in the page 10, since it appears early.

5) Fig. 3 will be better if both panels are presented on the same page.

6) Fig. 4 does not have panels, but 4 graphs are presented.

7) Fig. 5 does not have panels, but 4 graphs are presented.

8) Fig. 8 legend (p16.-17), it is not quite understood where words on p. 17 belong.

9) Are the bars are presented with SD or SE.

It is a plus for the manuscript that authors define all limitations of the approach at the end of the manuscript. It is a good piece of the work which is required further validations and exploration.

6. PLOS authors have the option to publish the peer review history of their article (what does this mean?). If published, this will include your full peer review and any attached files.

Reviewer #1: No

Reviewer #2: No

Reviewer #3: No

Reviewer #4: No

---

## [Author Response · Author response to Decision Letter 0]

6 Nov 2019

Reviewer #1: The idea of using the secretome of MSCs to treat an inflammatory disease is appropriate. However, the data do not support the conclusion. It appears that the MSC treatment was more superior to than the secretome. The condition media used a cut off that could consist of secretome other than cytokines. The CM used was from naive MSCs. It is possible when these cells are in an inflammatory environment the secretome will be different.

-> We thank the reviewer for their comments. The TMSC cells were cultured in low-glucose Dulbecco’s modified Eagle’s medium (DMEM) containing antibiotics and 10% heat-inactivated fetal bovine serum (FBS), until they attained a cell density of 5×105 cells/well. TMSC conditioned medium (TMSC-CM) was acquired after seeding 5×105 TMSC (in 10ml low-glucose DMEM containing 10% FBS) in a 100 mm dish and harvesting from the cells for 5–7 days (total 2×106 cells). Therefore, TMSC and TMSC-CM were both acquired from non-inflammatory environments. [Materials and Methods section; page 6-7].

The TMSC-CM group received 12 IP injections of 500 μL of TMSC-CM (equivalent to 1×105 TMSC) in order to match the amount of TMSC administered with the TMSC group. As the number of injections increases, the injection-related stress of the mouse increases. Therefore, the same number of injections was used to reduce the bias. Consequently, the triple-concentrated (using a 100 kDa cut-off centrifugal filter) TMSC-CM-conc group, which received four IP injections of 500 μL of TMSC-CM-conc (equivalent to 3×105 TMSC) was set up so that the number of injections would be equivalent to that of the TMSC group. [Materials and Methods section; page 7, 9].

In our results, DAI values, weight change, and recovery of colon length were improved in the TMSC, TMSC-CM and TMSC-CM-conc groups compared to that of the colitis control group. However, there were no significant differences between the DAI values, weight change and recovery of colon length for each of the three treatment groups. [Results section; page 15].

Reviewer #2: In the manuscript entitled “The Efficacy of Conditioned Medium Released by Tonsil-derived Mesenchymal Stem Cells in a Chronic Murine Colitis Model” the authors describe a new approach to treat chronic colitis by means of tonsil-derived mesenchymal stem cells (TMSC) in a mouse model of chronic colitis.

The authors have shown the efficacy of tonsil-derived mesenchymal stem cells conditioned medium (TMSC-CM) in the control of this inflammatory bowel disease clinical and immunological parameters but not of the histological colitis score.

The authors underline that the preparation and storing of TMSC-CM is clinically more feasible respect to TMSC isolation and suggest that TMSC-CM therapy could serve as a pharmacological agent to treat chronic colitis.

In my opinion these data are interesting and deserve publication on PlosOne. I suggest however that the authors discuss further the lack of histological score improvement and explain better the scheme of treatments in particular the higher quantity of injections in the TMSC-CM mouse group (12 respect to 4).

-> Thank you for your valuable comments.

1) We have now inserted the results of the histologic findings in the results section. [Results section; page 16].

“Histopathologic findings of the dissected colon specimen showed infiltration of mononuclear cells and crypt damage. Histologic colitis scoring, assessed after H&E staining, did not show any significant differences between each group (p = 0.2933, ANOVA) (Fig. 7).”

We have now discussed the reasons, as suggested, in the discussion section. [Discussion section; page 22].

“The scores obtained using the histological scoring system showed a significant difference between the treatment and colitis control groups in a previous study conducted using the acute colitis mouse model, in which colitis was induced for 7 days. However, in the current study using the chronic colitis model, in which colitis was induced for 30 days, it seems that the healing was not reflected in the scoring system because of the fibrotic change, resulting in no significant difference between the groups. Additional analysis regarding other histologic scoring systems should be performed. In addition, other staining methods such as Masson’s trichrome staining should be performed to evaluate the distribution of the collagen fibers, which is related to chronic inflammation and fibrosis.”

2) The TMSC-CM group received 12 IP injections of 500 μL of TMSC-CM (equivalent to 1×105 TMSC) in order to match the amount of TMSC administered with the TMSC group. As the number of injections increases, the injection-related stress of the mouse increases. Therefore, the same number of injections was used to reduce the bias. Consequently, the triple-concentrated (using a 100 kDa cut-off centrifugal filter) TMSC-CM-conc group, which received four IP injections of 500 μL of TMSC-CM-conc (equivalent to 3×105 TMSC) was set up so that the number of injections would be equivalent to that of the TMSC group. [Materials and Methods section; page 7, 9]. We have now modified figure 1, as requested. [Fig 1.]

Grammar and style should also be improved.

-> This work has now been further professionally edited for English content, as requested, and we attach the certificate.

Reviewer #3: Minor English wording and syntax changes are necessary. Otherwise, it is a very good paper an provides a ready means for MSC and their products in immunotherapy

-> We thank the reviewer for their comments. This work has now been further professionally edited for English content, as requested, and we attach the certificate.

Reviewer #4: The manuscript by Ko Eun Lee, Sung-Ae Jung et al. describes the development of the approach to reduce an inflammation in the chronic colitis using a conditioning media from a human tonsil-derived mesenchymal stem cells (TMSC). Authors are already published several papers regarding the injection of TMSCs in order to treat colitis in mice models. However, they found that TMSC cells are not migrated to the inflammation site in the intestine, therefore in the current manuscript they validated the conditioned media from the TMSCs.

They found that the CM can change the disease progression like cells itself.

Authors utilized a well-established mouse model using DSS (dextran sulfate sodium) to induce the chronic and acute colitis. The assessment of the severity disease is well-established, and authors employed these parameters in they work (weight of mice, the colon length, levels of proinflammatory cytokines).

Major specific comments:

1) The rate of splenocytes was reduced (p13), but this section causes the confusion. Proliferation rates was measured in what (days)? Fig. 2 presents the folds changes. Please clarify.

-> In the bar graph of Fig 2, the rate of proliferation of splenocytes stimulated by LPS only showed an increase in proliferation rate (fold change). However, when the splenocytes were co-cultured with TMSC or TMSC-CM in the presence of LPS, there was a decreased proliferation rate (fold change). [Results section; page 13].

2) It is understandable that the TMSC-CM causes the protective anti-inflammatory effect on the colonic inflammation. However, it is raising the question can this therapy impact on a mice immuno system causing the immunogenicity and other complications?

-> We thank the reviewer for their comments. When we evaluated the dissected colon specimen from sacrificed mice, they showed reduction of colon length and inflammation but no perforation. This information has now been added to the Results section. [Results section; page 16].

As this is the result of an in vitro study, it is possible that TMSC-CM may affect inflammatory and anti-inflammatory cytokines and therefore may alter the immune system. However, further in vitro study will be needed to evaluate the immunological mechanisms. [Results section; page 13-14].

3) How long after this treatment mice can survive compare with the disease bearing group?

-> In this study, we sacrificed mice with CO2 inhalation on day 31, so we did not assess how long the treated mice could survive for. The mice that died during the 30 day experiment died because of weight loss during the second inducing period of 1.5% DSS. After that period, the mice generally survived. This information has now been added to the Materials and Methods section. [Methods section; page 10].

4) p.15 Fig. 4, Disease activity index (DAI) was define in the legend, please do it in the page 10, since it appears early.

-> “Disease activity index (DAI)” was mentioned and defined in the ‘Introduction section’ (page 5). Therefore, in accordance with the journal guidelines, further description was not needed in the Method section (page 10). [Introduction section; page 5].

5) Fig. 3 will be better if both panels are presented on the same page.

-> As suggested, panel B has now been moved to underneath panel A, in Fig. 3.

6) Fig. 4 does not have panels, but 4 graphs are presented.

-> We have now inserted the panel names of (A), (B), (C) and (D), and modified part of Fig. 4.

7) Fig. 5 does not have panels, but 4 graphs are presented.

-> We have now inserted the panel names of (A), (B), (C) and (D), and modified part of Fig. 5.

8) Fig. 8 legend (p16.-17), it is not quite understood where words on p. 17 belong.

-> We have now changed the location of the legend to Fig. 8. This is now displayed on one side of the paper.

9) Are the bars are presented with SD or SE.

-> The error bars of all graphs show standard deviation (SD), and this description has now been added to all figure legends.

It is a plus for the manuscript that authors define all limitations of the approach at the end of the manuscript. It is a good piece of the work which is required further validations and exploration.

---

## [Editor Report · Decision Letter 1]

12 Nov 2019

The Efficacy of Conditioned Medium Released by Tonsil-derived Mesenchymal Stem Cells in a Chronic Murine Colitis Model

PONE-D-19-18142R1

Dear Dr. Jung,

We are pleased to inform you that your manuscript has been judged scientifically suitable for publication and will be formally accepted for publication once it complies with all outstanding technical requirements.

With kind regards,

Debabrata Banerjee, Ph.D

Academic Editor

PLOS ONE

Additional Editor Comments (optional):

The revised version of the manuscript is acceptable. It will be of interest to readers of PLOsOne
---

## [Editor Report · Acceptance letter]

20 Nov 2019

PONE-D-19-18142R1 

The Efficacy of Conditioned Medium Released by Tonsil-derived Mesenchymal Stem Cells in a Chronic Murine Colitis Model 

Dear Dr. Jung:

I am pleased to inform you that your manuscript has been deemed suitable for publication in PLOS ONE. Congratulations! Your manuscript is now with our production department. 

With kind regards,

on behalf of

Dr. Debabrata Banerjee 

Academic Editor

PLOS ONE